# Substrate-Specific Activation of α-Secretase by 7-Deoxy-Trans-Dihydronarciclasine Increases Non-Amyloidogenic Processing of β-Amyloid Protein Precursor

**DOI:** 10.3390/molecules25030646

**Published:** 2020-02-03

**Authors:** Yoon Sun Chun, Yoon Young Cho, Oh Hoon Kwon, Dong Zhao, Hyun Ok Yang, Sungkwon Chung

**Affiliations:** 1Natural Products Research Center, Korea Institute of Science and Technology, Gangneung 25451, Korea; ysun129@skku.edu (Y.S.C.); 614003@kist.re.kr (D.Z.); 2Department of Physiology, Sungkyunkwan University School of Medicine, Suwon 16419, Korea; myjubilate@hotmail.co.kr (Y.Y.C.); drummer0114@naver.com (O.H.K.); 3Division of Bio-Medical Science & Technology, KIST School, Korea University of Science and Technology, Seoul 02792, Korea

**Keywords:** β-amyloid, Alzheimer’s disease, β-amyloid precursor protein, α-secretase, 7-deoxy-trans-dihydronarciclasine, *Lycoris chejuensis*

## Abstract

Accumulation of β-amyloid (Aβ) in the brain has been implicated in the pathology of Alzheimer’s disease (AD). Aβ is produced from the Aβ precursor protein (APP) through the amyloidogenic pathway by β-, and γ-secretase. Alternatively, APP can be cleaved by α-, and γ-secretase, precluding the production of Aβ. Thus, stimulating α-secretase mediated APP processing is considered a therapeutic option not only for decreasing Aβ production but for increasing neuroprotective sAPPα. We have previously reported that 7-deoxy-trans-dihydronarciclasine (E144), the active component of *Lycoris chejuensis*, decreases Aβ production by attenuating APP level, and retarding APP maturation. It can also improve cognitive function in the AD model mouse. In this study, we further analyzed the activating effect of E144 on α-secretase. Treatment of E144 increased sAPPα, but decreased β-secretase products from HeLa cells stably transfected with APP. E144 directly activated ADAM10 and ADAM17 in a substrate-specific manner both in cell-based and in cell-free assays. The Lineweaver–Burk plot analysis revealed that E144 enhanced the affinities of A Disintegrin and Metalloproteinases (ADAMs) towards the substrate. Consistent with this result, immunoprecipitation analysis showed that interactions of APP with ADAM10 and ADAM17 were increased by E144. Our results indicate that E144 might be a novel agent for AD treatment as a substrate-specific activator of α-secretase.

## 1. Introduction

Alzheimer’s disease (AD) is characterized by the accumulation of extracellular senile amyloid plaques and intracellular neurofibrillary tangles composed of hyper-phosphorylated tau [1,2]. Hyperphosphorylated tau aggregates into paired helical filaments, which then form the neurofibrillary tangles and affect neurons. Thus, one strategy in AD treatment is tau-targeting [3]. Neurotoxic β-amyloid (Aβ) peptides are the dominant components of senile plaques that play a central role in the progression of AD. The aggregation of Aβ provokes neuronal dysfunction, memory impairment, and synaptic damage [4,5]. Aβ is derived from the Aβ precursor protein (APP) processed by β-secretase (BACE1) and γ-secretase. In the initial step, APP can be cleaved by β-secretase producing soluble ectodomain fragment (sAPPβ) and C-terminal fragment (CTF), C99. C99 is further cleaved by γ-secretase to yield several Aβ species with different length and APP intracellular domain (AICD). γ-Secretase complex firstly cleaves C99 into longer Aβ peptides, Aβ48 or Aβ49. Then, it further trims longer Aβ peptides into a shorter variety of Aβ peptides, from Aβ38 to Aβ43, depending on its cleavage sites [6,7]. This amyloidogenic pathway mainly releases Aβ40 and Aβ42. Even though Aβ40 is the most abundant form, Aβ42 is more prone to oligomerization and aggregation, which makes Aβ42 a major pathogenic factor in AD [5,6]. Alternatively, APP can be cleaved by α-secretase within the center of the Aβ domain producing another soluble ectodomain fragment (sAPPα) and C-terminal fragment (C83) precluding the Aβ generation. C83 is further cleaved by γ-secretase to generate p3 peptide and AICD. This is known as the non-amyloidogenic pathway [8,9].

Three of A Disintegrin and Metalloproteinases (ADAM) family have been suggested as potential α-secretase: ADAM9, ADAM10, and ADAM17/tumor necrosis factor-α (TNFα) converting enzyme (TACE) [10,11]. α-Secretase can be a constitutive cleavage enzyme or be stimulated by several G protein-coupled receptors (GPCRs) activating drugs and several kinases such as protein kinase C (PKC), phosphatidyl-inositol 3-kinase, and mitogen-activated protein kinase [12,13]. Among three family members, ADAM10 is considered as sheddase for APP and other diverse cell-surface proteins, including cytokines, cell adhesion molecules (CAMs), and Notch [13,14]. In contrast, ADAM9 and ADAM17 are believed to undertake regulated APP cleavage [13,15,16]. However, several studies demonstrated that ADAM9 could mimic the constitutive function of ADAM10 in microglia cells [17,18]. In addition to APP cleavage, ADAM17 is involved in ectodomain shedding of surface-bound molecules, including TNFα and EGF receptor ligands as well as shedding over 80 substrates [19,20,21,22]. In addition, ADAM17 contributes to the myelination, nerite outgrowth [15] and AD-related neuroinflammation [23]. 

Several reports have shown that enhancing α-secretase activity might be a potential strategy in AD treatment. The level of sAPPα in the cerebrospinal fluid of the AD patients has been found to be significantly decreased, indicating that α-secretase activity is reduced in AD [24,25]. Overexpression of ADAM10 decreases the formation of senile plaques but increases the secretion of sAPPα in an animal model of AD, while overexpression of the ADAM10 inactive mutant form increases the formation of senile plaques [26]. Mutation of ADAM10 (Q170H and R181G) has been found in AD patients [27]. Furthermore, it is reported that ADAM17-positive neurons are localized adjacent to the amyloid plaques in AD brains [28]. Overexpression of ADAM17 in HEK293 cells increased sAPPα release [29]. Inhibition and activation of ADAM17 regulated secretion of sAPPα in vitro and in vivo [30]. Thus, stimulating α-secretase mediated APP processing is a therapeutic option for the treatment of amyloid pathology by decreasing Aβ production. There have been intensive efforts to find small molecule ADAM activators, and several activators for ADAM are demonstrated [14]. However, this strategy should consider proteolysis of APP as well as various substrates, including TNFα and epidermal growth factor receptor ligands involved in inflammation and cancer [13]. Especially, ADAM17 participates in the shedding of diverse inflammatory factors [31]. For these reasons, a small molecule ADAM activator for AD treatment should be specific for APP sparing other substrates to avoid many deleterious consequences.

We have shown the effect of *Lycoris chejuensis* K. Tae et S. Ko (CJ) originated from Jeju Island in Korea on Aβ generation and spatial memory ability both in vivo and in vitro [32]. Recently, we have identified 7-Deoxy-trans-dihydronarciclasine (Figure 1; coded as E144) as the active component of CJ [33]. In this study, we further examined the effect of E144 on Aβ production. Acute treatment with E144 increased sAPPα secretion and CTFα level but decreased CTFβ and Aβ levels. Using a cell-free assay, we found that E144 directly activated ADAM10 and ADAM17 in a substrate-specific manner. Lineweaver–Burk plot analysis revealed that E144 enhanced the affinity of ADAM17 towards its substrate. Consistent with this result, E144 increased the interaction of APP with ADAM10 and ADAM17. These results suggest that E144 can increase non-amyloidogenic processing of APP by activating ADAM10 and ADAM17.

## 2. Results

### 2.1. E144 Increases Secreted sAPPα Level but Decreases Aβ Levels

We tested the effect of E144 on sAPPα production from HeLa cells stably transfected with APP carrying Swedish mutation (APPsw). Cells were incubated with 1 μM E144 for 1, 2, 5, or 8 h. Levels of sAPPα in conditioned media were then measured using a specific ELISA kit (Figure 2a). When cells were incubated with E144 for 1 h, the level of sAPPα was significantly increased by 29.7% ± 8.4% (*n* = 6). The level of sAPPβ was decreased by E144, although the effect was not significant (2.3% ± 8.4%, *n* = 6). The minimal effect of E144 on sAPPβ might be explained by the preferential APPsw cleave by β-secretase over α-secretase [34]. These results also indicated that the effect of E144 on sAPPα level was not due to changed APP transport to the membrane. However, after more than 2 h incubation, the levels of sAPPα and sAPPβ were decreased by E144 in a time-dependent manner. This might be because E144 decreases APP levels, as we have previously shown using Western blots [33]. We reported that the levels of total, mature, and immature APP were decreased by E144 in a time-dependent manner. These results indicated that E144 increased the secretion of sAPPα with 1 h of treatment time. We also tested the secreted level of sAPPα using Western blot. Cells were incubated with 1 μM E144 for 1, 2, or 8 h. Conditioned media were then concentrated and immunoprecipitated. As shown in Figure 2b, the level of sAPPα was significantly increased by more than 2-fold after 1 h incubation with 1 μM E144 (*n* = 5). However, the level of sAPPα was significantly decreased at 8 h after incubation with E144. Apparently, the effect of E144 on sAPPα seemed much larger when we used the Western blot than when we used ELISA. This could be because the conditioned media were concentrated and immunoprecipitated using APP antibody for Western blot. We also tested the effects of E144 on human neuroblastoma SH-SY5Y cells, stably transfected with wild type APP. Even though Aβ42 levels were too low to detect, levels of sAPPα in conditioned media were significantly increased by E144 after 1 h incubation (Appendix A). The level of sAPPβ was not changed by E144.

We then measured levels of CTFα and CTFβ, representing the products of α- and β-secretase, respectively. Cells were incubated with 0.5, 1, or 5 μM E144 for 1 h, and levels of CTF in cell lysates were analyzed using Western blot. A typical Western blot result is shown in Figure 2c. Relative band densities of CTF are shown in Figure 2d (*n* = 4). E144 increased the level of CTFα but decreased the level of CTFβ. These results indicated that E144 increased non-amyloidogenic processing of APP through α-secretase but decreased amyloidogenic processing through β-secretase. Since E144 decreased the amyloidogenic processing of APP, we examined the effect of E144 on Aβ levels. Cells were incubated with 0.5, 1, or 5 μM E144 for 1 h, and levels of Aβ42 and Aβ40 in conditioned media were measured using specific ELISA kits. The level of Aβ42 was significantly decreased by 13.6% ± 3.5%, 18.0% ± 2.1% and 24.6% ± 4% after incubation with 0.5, 1, and 5 μM E144, respectively (Figure 2e; *n* = 4). The level of Aβ40 was also significantly reduced by 9.8 ± 1.1%, 15.2% ± 3.6% and 25.9% ± 5.3% after incubation with 0.5, 1, and 5 μM E144, respectively (Figure 2f; *n* = 4).

### 2.2. E144 Directly Activates ADAM17

Since E144 increased non-amyloidogenic processing of APP by α-secretase, we investigated whether E144 affected α-secretase activity. First, we performed a cell-based assay for ADAM17 activity using APPsw-transfected HeLa cells. The cell lysate was added into an anti-human ADAM17-coated plate and incubated for 1 h to capture ADAM17. Then, Mca-KPLGL-Dpa-AR, a fluorogenic ADAM17 substrate, was added and incubated for an additional 5 h with 0.1, 0.5, 1, or 5 μM E144. Fluorescence of the quenched substrate was directly related to the enzyme activity. TAPI-1, an ADAM17 inhibitor, inhibited the activity of ADAM17 by 83.1% (*n* = 6), confirming that ADAM17 activity could be measured from cell lysates (Figure 3a). E144 at 0.5, and 1 μM significantly increased ADAM17 activity by 32.1% and 70.8%, respectively (*n* = 6). ADAM17 activity was increased by more than 4-fold after incubation with 5 μM E144. Thus, E144 activated ADAM17 in a concentration-dependent manner. We also performed a cell-free assay for ADAM17 activity using human recombinant ADAM17. E144 increased the activity of human recombinant ADAM17, as shown in Figure 3b (*n* = 4). ADAM17 activity was markedly increased by 2-fold after incubation with 5 μM E144. This result using recombinant ADAM17 in a cell-free environment suggested that E144 directly activated ADAM17.

E144 at 1 or 5 μM E144 increased the activity of human recombinant ADAM17 in a time-dependent manner when we used a fluorogenic ADAM17 substrate (Appendix A). We then tested whether the activating effect of E144 on ADAM17 was blockable by the presence of an ADAM17 inhibitor. Even in the presence of TAPI-1, E144 was able to activate ADAM17 with similar potency as in the absence of TAPI-1 (Appendix A). Thus, the activating effect of E144 on ADAM17 was independent of TAPI-1, the known inhibitor of ADAM17.

We examined the substrate-dependent effects of 1 or 5 μM E144, as shown in Figure 3c. Human recombinant ADAM17 was incubated with 0.02, 0.4, 1, 2, 4, or 10 mM Mca-KPLGL-Dpa-AR. Lineweaver–Burk plot analysis was conducted to define the effect of E144 on ADAM17-catalyzed reaction (Figure 3d; *n* = 4). When the fluorescence intensities were plotted on the X-axis, and the substrate concentrations were plotted on the Y-axis, they showed different slopes that intersected together at an identical point. The V_max_ of the enzyme was 26.2 mM/min and K_m_ was 0.02 mM in the presence of 5 μM E144, compared with the V_max_ of 11.4 mM/min and K_m_ of 0.58 mM for the control. Thus, E144 decreased Michaelis–Menten constant K_m_ while maximal velocity, V_max_, was apparently not or moderately affected. 

### 2.3. E144 Activates ADAM17 in a Substrate-Specific Manner

It is known that ADAM17 mediates the cleavage of the substrates involved in brain pathology, inflammation, and cancer [35]. TNFα, one of the cardinal pro-inflammatory cytokines, is a well-known substrate for ADAM17 [20,21]. Thus, activation of ADAM17 by E144 might increase TNFα secretion and lead to an inflammatory reaction. To test this possibility, we measured secreted levels of TNFα from microglial BV-2 cells. The secreted TNFα level without lipopolysaccharide (LPS) stimulation in these cells was undetectable with the ELISA method. Thus, cells were pre-incubated with 1 μg/mL LPS for 3 h, and the culture medium was exchanged with a fresh one. Cells were then treated with 0.5 or 1 μM E144 in the presence of LPS for an additional 1 h. TNFα levels in conditioned media were measured using a specific ELISA kit (Figure 4a). The secretion of TNFα was significantly increased by stimulation with LPS. However, the presence of 1 μM E144 decreased LPS-induced TNFα level by 14.7% (*n* = 5). These results indicated that while E144 activated ADAM17 towards APP and a fluorescent substrate, Mca-KPLGL-Dpa-AR, it also inhibited ADAM17 towards another substrate, thereby reducing the secreted TNFα. To clarify this substrate-specific activating effect of E144 on ADAM17, we measured ADAM17 activity using another fluorogenic ADAM17 substrate, QXL520/5-FAM. TAPI-1, an ADAM17 inhibitor, inhibited the activity of human recombinant ADAM17 (Figure 4b; *n* = 5). When recombinant ADAM17 was incubated with 1 or 5 μM E144 for 40 min, ADAM17 activity did not change. Considering these results, the activating effect of E144 was specific for certain substrates. 

### 2.4. E144 Activates ADAM10 in a Substrate-Specific Manner

We next tested the effect of E144 on ADAM10 activity in both a cell-based and cell-free assay. For the cell-based assay, ADAM10 from the lysate of APPsw-transfected HeLa cells was incubated with 1 or 5 μM E144 for 40 min in the presence of the ADAM10 substrate, 5-FAM/QXL520. Fluorescence of the quenched substrate was directly related to the enzyme activity. ADAM10 activity did not change by 1 and 5 μM of E144 (Figure 5a, *n* = 4). E144 did not affect human recombinant ADAM10 in a cell-free activity assay either (Figure 5b, *n* = 4). The inhibitor of ADAM10, GM-6001, inhibited ADAM10 activity by 80%. We next tested the effect of E144 on human recombinant ADAM10 activity using another substrate, Mca-KPLGL-Dpa-AR, the same substrate used in Figure 3a,b for the ADAM17 activity assay. GM-6001 inhibited the activity of ADAM10 by 35% (Figure 5c). When ADAM10 was incubated with Mca-KPLGL-Dpa-AR for 1 h, E144 activated ADAM10 activity in a concentration-dependent manner. E144 at 5 μM concentration significantly increased ADAM10 activity by 27.9% (*n* = 4). These results indicated that E144 could activate ADAM10 and ADAM17 in a substrate-specific manner. However, the activating effect of E144 on ADAM10 was less potent than that on ADAM17 although we used the same substrate, Mca-KPLGL-Dpa-AR.

ADAM10 is the major physiological α-secretase mediating the cleavage of APP in the neurons. In contrast, ADAM9 and ADAM17 participate in a regulated cleavage of APP [15]. Since we observed that the activating effect of E144 on ADAM17 was more potent than that on ADAM10, we tried to identify the target of E144 in physiological condition with intact cells. We used specific siRNAs for ADAM9, 10, and 17. As shown in Figure 6a, we were able to reduce expression levels of these proteins by more than 90% using specific siRNAs. When ADAM9 or ADAM10 was down-regulated, E144 was able to decrease Aβ42 levels similar to control siRNA-treated cells (Figure 6b; *n* = 6). In contrast, the effect of E144 was completely prevented when ADAM17 was down-regulated. These results suggest that the effect of E144 on Aβ might be mostly via ADAM17 in intact cells. 

### 2.5. E144 Increases Interaction of APP with ADAM10 and ADAM17

We showed that E144 directly activated ADAM10 and ADAM17 using recombinant enzymes. We also showed that E144 enhanced the affinity ADAM17 towards the substrate. In addition, APP can bind to the α-/β-secretase binary complex [36]. Thus, we tested the effect of E144 on the interaction of APP with ADAM10 and ADAM17 (ADAMs) by performing co-immunoprecipitation analysis in APPsw-transfected HeLa cells. Cells were treated with 1 or 5 μM E144 for 1 h followed by immunoprecipitation of cell lysate with APP antibody and probing for ADAMs using Western blot. A typical result is shown in the *upper panel* of Figure 7a. Relative band densities of immunoprecipitated ADAM9, ADAM10, and ADAM17 are shown in Figure 7b (*n* = 4). E144 did not affect the interaction between APP and ADAM9. In contrast, E144 increased both APP-ADAM10 and APP-ADAM17 interactions. We next measured the expression level of ADAMs. Cells were treated with 1 or 5 μM E144 for 1 h. Levels of ADAMs in cell lysates were then analyzed using Western blot. A typical Western blot result is shown in the *lower panel* of Figure 7a. The relative band densities of ADAMs compared to β-tubulin are shown in Figure 7c (*n* = 4). ADAMs exist as a pro-enzyme in the trans-Golgi network and undergo maturation through removal of pro-domain, thereby converting to a mature, active form of the enzymes [37]. Levels of both precursor and active ADAM9 were significantly decreased by 5 μM E144. However, levels of ADAM10 or ADAM17 were not significantly changed by E144. These results indicate that activating effects of E144 on ADAM10 and ADAM17 might be due to increased interaction between APP and ADAMs by E144.

### 2.6. Differential Effects of Structural Analogues of E144 on ADAM10 and ADAM17

Previously, we have confirmed that three active components of CJ, E144, Narciclasine (E2), and 7-Deoxynarciclasine (E3), can decrease the production of Aβ by attenuating APP levels [32]. Since E2 and E3 were structural analogs of E144, we tested whether E2 and E3 could affect the activities of ADAM10 and ADAM17. When human recombinant ADAM10 was used, E2 increased ADAM10 activity like E144 (Figure 8a). However, E3 did not change the ADAM10 activity. Similarly, E2, but not E3, increased ADAM17 activity when we used cell lysates (Figure 8b). Thus, structural analogs of E144 showed differential effects on ADAM10 and ADAM17 activities.

## 3. Discussion

We have recently reported that E144 can attenuate the secretion of Aβ through decreasing the APP level and delaying the maturation of APP [33]. Notably, chronic treatment with E144 significantly decreased levels Aβ and APP in brains of Tg2576 mice, an APP transgenic mouse model of AD. E144 also rescued behavioral deficits in TG mice. In the present study, we further found that acute treatment with E144 caused direct activation of α-secretase in HeLa cells transfected with APPsw. Consequently, E144 promoted the non-amyloidogenic processing of APP, thereby increasing levels of sAPPα and CTFα while decreasing levels of CTFβ and Aβ. E144 mostly decreased K_m_ of ADAM17. This may indicate that E144 enhances the affinity of ADAM17 towards its substrates. It is possible that E144 can modulate interactions between α-secretase and APP, affecting APP cleavage and Aβ generation. Consistent with this result, E144 increased the interaction of APP with ADAM10 and ADAM17 when we performed co-immunoprecipitation analysis. E2, a structural analog of E144, also increased ADAM10 and ADAL17 activities. Interestingly, however, another analog, E3, did not change the activities of these enzymes. 

One strategy for AD therapeutics is stimulating α-secretase to inhibit the amyloidogenic processing of APP. sAPPα, a product from APP by α-secretase, has been shown to be beneficial for memory function, to have neuroprotective properties, and to be able to stabilize neuronal calcium homeostasis [38,39,40,41,42]. We found that E144 had a beneficial effect in upregulating the secretion of sAPPα by acute treatment. However, the level of sAPPα was decreased by E144 in a time-dependent manner at more than 2 h after treatment. This event is likely to be attributable to the decrease in APP level by E144 [33]. Our data using fluorogenic peptide substrates showed that E144 directly activated ADAM17 in a cell-free assay. In addition, we showed that E144 was a substrate-specific ADAM17 activator. Even though fluorogenic peptides used in our experiments were designed to reduce the cross-reactivities between ADAMs, further studies using full-length cellular substrates will be needed to confirm the substrate-specificity of E144. In addition, further experiments are needed to elucidate how E144 activates ADAM10 and ADAM17 in a substrate-specific manner.

ADAM proteins can cleave various substrates, including TNF-α, an epidermal growth factor involved in inflammation and cancer [43,44]. Recent studies have implicated that activation of α-secretase can induce unfavorable effects because ADAM proteins can increase the processing of other substrates, resulting in the promotion of tumorigenesis, tumor growth, and inflammation [13]. Thus, the importance of substrate-specific ADAM targeting has been highlighted in the AD research field to avoid side-effect [14]. Our data showed that E144 decreased the secreted level of TNF-α, ruling out the possibility that E144 might cause putative side-effects by increasing TNF-α. Instead, E144 significantly decreased LPS-induced TNFα level, consistent with a recent report regarding the anti-inflammatory effect of E144 [45]. These results indicated that while E144 activated ADAM17 towards certain substrate such as Mca-KPLGL-Dpa-AR, it also inhibited ADAM17 towards another substrate, thereby reducing the secreted TNFα.

Some natural compounds, also extracted from plants, have been studied to activate α-secretase for treating AD. Resveratrol, a natural polyphenolic flavonoid extracted from several plants such as grapes, has neuroprotective and antioxidant properties [46]. It can also promote the level of ADAM10 expression, which may induce an increase of CTFα level in APP-transfected CHO cells [47]. Curcumin isolated from the plant *Curcuma longa* has antioxidant and anti-inflammatory properties [48]. Curcumin amino-acids conjugates with isoleucine, phenylalanine, or valine enhance the expression level of ADAM10 and secretion of sAPPα [49]. In line with that, a recent study revealed that melatonin enhanced non-amyloidogenic processing inhibited Aβ aggregation and amyloid plaques via increasing ADAM10 expression and its activity [14]. Retinoic acid was also suggested as a transcriptional activator for ADAM10 increasing ADAM10 expression, consequently decreasing Aβ production [50]. Furthermore, it has been reported that acitretin, a second-generation retinoid, enhances the up-regulation of α-cleavage processing of APP accompanying the increase of sAPPα levels in human patient CSF [51]. Importantly, the study of Brummer and colleagues showed that acitretin has the substrate-specific ADAM10 activating effect [52].

Our results indicated that E144 and E2, natural compounds isolated from the plant, could be used as a potential agent for the treatment of AD by activating ADAM10 and ADAM17 in a substrate-specific manner. However, further experiments have to include various substrates of ADAM10 and ADAM17 to avoid many deleterious consequences by activating these enzymes. Moreover, additional experiments are needed to clearly show that the major target of E144 is ADAM17 over ADAM10 since it has been reported that ADAM17-mediated alpha cleavage of APP, referred to as the “regulated” pathway, is mostly occurring when PKC is stimulated [13,15,16]. Since activation of GPCR can increase APP processing by α-secretase via ADAM17 [13], the effect of E144 on ADAM17 may occur through GPCR signaling. It is also possible that the components present in the cell enhance the action of E144, increasing the affinity of ADAM17 toward its substrate. This may also explain why recombinant ADAM17 exhibited a lower level of activation by E144 (Figure 3a versus Figure 3b). Several binding partners of ADAM17 have been identified to enhance ADAM17 activity. Thioredoxin-1 is involved in the regulation of ADAM17 activity as a partner of the ADAM17 cytoplasmic domain in HEK293 cells [53]. The scaffolding protein synapse associated protein 97 as a binding partner of the cytoplasmic domain of TACE in mammalian cells regulates TACE shedding activity [54]. Thus, binding partners and scaffolding protein of ADAM17 could increase the action of E144, enhancing the affinity between ADAM17 and its substrate in the cellular environment.

## 4. Materials and Methods

### 4.1. Cell Culture and Experimental Treatments

HeLa cells stably transfected with an APP carrying Swedish mutation (APPsw) were cultured at 37 °C with 5% CO_2_. Dulbecco’s Modified Eagle Medium (DMEM) was used with 10% heat-inactivated fetal bovine serum (FBS) containing 100 units/mL penicillin, 100 μg/mL streptomycin, 260 μg/mL Zeocin, and 400 μg/mL G418. BV-2 microglia cells were cultured at 37 °C with 5% CO_2_ in F12/DMEM (1:1) with 10% FBS, 100 units/mL penicillin, and 100 μg/mL streptomycin. LPS (Sigma, 62326, St. Louis, MO, USA) was dissolved in water.

### 4.2. sAPPα Immunoprecipitation

APPsw-transfected HeLa cells were incubated with 6 mL culture media containing DMSO or E144 at a density of 5 × 10^6^ cells in 100 mm dish. Two 100 mm dishes were prepared for each condition. Conditioned media of 12 mL volume for each condition were collected and concentrated to 100 μL using a kit (Millipore, ACK5030GS). Concentrated media were immunoprecipitated with Protein G Agarose (Millipore). APP antibody against N-terminus (Abcam, Cambridge, UK) was used. For the Western blot, immunoprecipitated samples were washed 3 times with PBS and probed for sAPPα (Covance, Princeton, NJ, USA).

### 4.3. Protein Extraction and Western Blotting

APPsw-transfected HeLa cells were washed with PBS and homogenized using lysis buffer (100 mM NaCl, 1% Triton X-100,1 mM sodium orthovanadate, protease inhibitors, 50 mM HEPES, pH 7.2). Centrifugation of cell lysates was carried out at 10,000× *g* for 10 min. The Bradford assay (Bio-Rad, Hercules, CA, USA) was used to measure protein concentration in the supernatant. SDS-PAGE (8% or 15%) was used to resolve proteins. The resolved proteins were transferred to a polyvinylidene fluoride membrane. Then, it was blocked using nonfat milk powder (5%) in Tris-buffered saline/Tween 20 (TBST) for 1 h at room temperature. The membrane was then incubated overnight at 4 °C with anti-sAPPα (Covance), anti-CTF (Sigma), anti-ADAM9 (Cell Signaling Technology, Danvers, MA, USA), anti-ADAM10 (Abcam), anti-ADAM17 (Abcam), anti-β-tubulin (Sigma), or β-actin (EnoGene, New York, USA). After washing membranes using TBST, horseradish peroxidase-conjugated goat anti-mouse IgG or goat anti-rabbit IgG (Cell Signaling Technology) were incubated at room temperature for 1 h. Enhanced chemiluminescence was performed to visualize the peroxidase activity. Multi Gauge Software (Fujifilm, Tokyo, Japan) using a LAS-4000 system was used to quantify the detected signals.

### 4.4. sAPPα and Aβ Peptide Assay

APPsw-transfected HeLa cells were treated with chemicals dissolved in DMSO. Specific ELISAs were used to measure levels of sAPPα (IBL), Aβ42 (Invitrogen, Carlsbad, CA, USA), and Aβ40 (Invitrogen) from conditioned medium according to the supplier’s instructions.

### 4.5. TNFα Assay

BV-2 cells were treated with 1 μg/mL LPS (Sigma) for 3 h and then cultured with E144 for 1 h at 37 °C. The cultured medium was collected, and the level of TNFα was measured using specific ELISA (R&D Systems) according to the supplier’s instruction.

### 4.6. α-Secretase Activity Assay

To measure ADAM17 activity for a cell-based assay using the TACE activity kit (Calbiochem, CBA042, San Diego, CA, USA), APPsw-transfected HeLa cells were collected and centrifuged at 700× *g* for 5 min. After washing with PBS, cells were suspended in extraction buffer (Novagen, Madison, WI, USA) at a final density of 2 × 10^7^ cells/mL. Lysates were incubated on ice for 30 min and centrifuged at 10,000× *g* for 5 min. The supernatant was diluted 1:1 with a sample buffer. TAPI-1 (Calbiochem) was dissolved in DMSO. TACE activity was then measured according to the supplier’s instructions using a fluorogenic ADAM17 substrate, Mca-KPLGL-Dpa-AR-NH_2_. After incubation at 37 °C for 5 h, fluorescence was measured at an excitation wavelength of 320 nm and an emission wavelength of 400 nm. Human recombinant ADAM17 was used to measure ADAM17 activity for a cell-free assay. It was diluted with sample buffer to yield a final concentration of 200 ng/mL. The Lineweaver–Burk plot was obtained using a fluorogenic ADAM17 substrate and recombinant ADAM17. Recombinant ADAM17 was incubated with a ADAM17 substrate at 0.02, 0.4, 1, 2, 4, or 10 mM in the presence or absence of E144. The change in fluorescence intensity was plotted on the Y-axis, and substrate concentration was plotted on the X-axis. ADAM17 activity was also measured using another TACE activity kit (AnaSpec, AS-72085, Fremont, CA, USA). Recombinant ADAM17 (200 ng/mL) was incubated with a different fluorogenic ADAM17 substrate, QXL520/5-FAM, in the presence or absence of E144. Fluorescence intensity was measured at an excitation wavelength of 490 nm and an emission wavelength of 520 nm.

To measure ADAM10 activity for a cell-based assay, cells were collected and washed with PBS. Next, cells were suspended in cold assay buffer at a final density of 8 × 10^6^ cells/mL. Lysates were incubated on ice for 30 min and centrifuged at 10,000× *g* for 5 min. ADAM10 activity from the supernatant was analyzed using a kit (AnaSpec, AS-72226) and the substrate, FAM/QXL520, according to the supplier’s instructions. Human recombinant ADAM10 was used for a cell-free assay. It was diluted with an assay buffer to yield a final concentration of 250 ng/mL. The substrate was incubated in the presence of E144 for 40 min at 37 °C. Fluorescence was measured at an excitation wavelength of 490 nm and an emission wavelength of 520 nm. The relative fluorescence intensity units were calculated compared to the control.

### 4.7. siRNA-Mediated Knockdown of ADAM9, ADAM10, and ADAM17

Specific siRNA oligonucleotides against ADAM9, ADAM10, and ADAM17 were designed as follows: ADAM9, 5′-GCTGGGAGGCTCTTCCTCTTCATCCTT-3′, ADAM10, 5′-CGTACACACAATAAACCTT-3, and ADAM17, 5′-AACAAATCTCCAAAGTGGCTCTATGTT-3′. Negative control oligonucleotides (SN-1001) were purchased from Bioneer (Oakland, CA, USA).

To knockdown ADAMs, the APPsw-transfected HeLa cells were pre-incubated in a serum-free medium without antibiotics for 1 h, followed by incubation in serum- and antibiotic-free medium with 20 nM siRNA duplexes using Lipofectamine 2000 (Invitrogen) according to the manufacturer’s instruction. After 4 h, FBS was added to make 10% FBS in medium (*v*/*v*). At 24 h post-siRNA treatment, antibiotic-free media with 10 nM siRNA duplexes using Lipofectamine 2000 were added and incubated for an additional 48 h.

To determine the effects of E144 on ADAMs, cells were washed with pre-warmed PBS to remove siRNA duplexes after 72 h of treatment with siRNAs. Cells were then incubated with complete medium containing 1 μM E144. At 1 h post-treatment with E144, cells were harvested with APP lysis buffer (50 mM HEPES, pH 7.2, 100 mM NaCl, 1% Triton X-100, and 1 mM sodium orthovanadate, and protease inhibitors). The supernatant was stored at −20 °C until measuring levels of secreted Aβ42 using Human Aβ42 ELISA High sensitivity Kit (Millipore, Burlington, VT, USA). The same amount of proteins from total lysates was subjected to Western blot analysis to determine the efficiency of siRNA mediated knockdown of ADAMs.

### 4.8. Co-Immunoprecipitation

APPsw-transfected HeLa cells were incubated with E144 for 1 h at 37 °C. Cells were washed with PBS and homogenized using a lysis buffer. Cell lysates were centrifuged at 10,000× *g* for 10 min at 4 °C. Then, the supernatant was collected. Total proteins (300 μg) were incubated with 2 μL antibody directed against APP (6E10; Covance) for 2 h. Immunoprecipitated proteins were washed 3 times with PBS. Western blot was then performed for analysis.

### 4.9. Statistical Analysis

Mean ± SEM was used to express all data. Statistical comparisons between controls and treated experimental groups were analyzed using Student’s *t*-test. *p* < 0.05 was considered statistically significant.

## Figures and Tables

**Figure 1 molecules-25-00646-f001:**
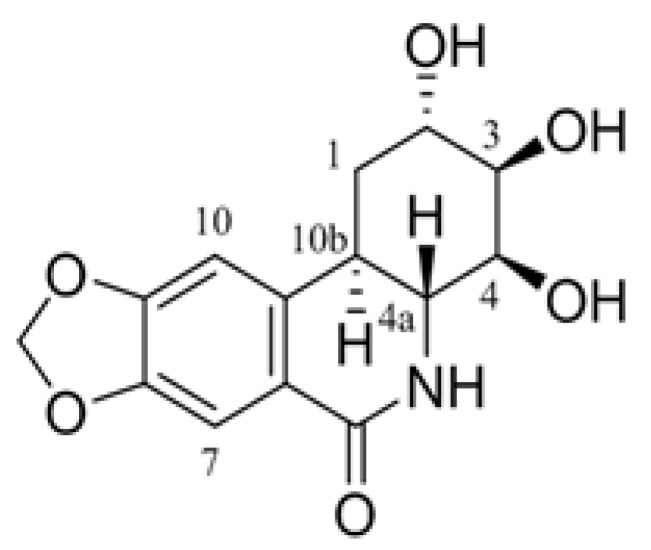
Chemical structure of 7-Deoxy-trans-dihydronarciclasine.

**Figure 2 molecules-25-00646-f002:**
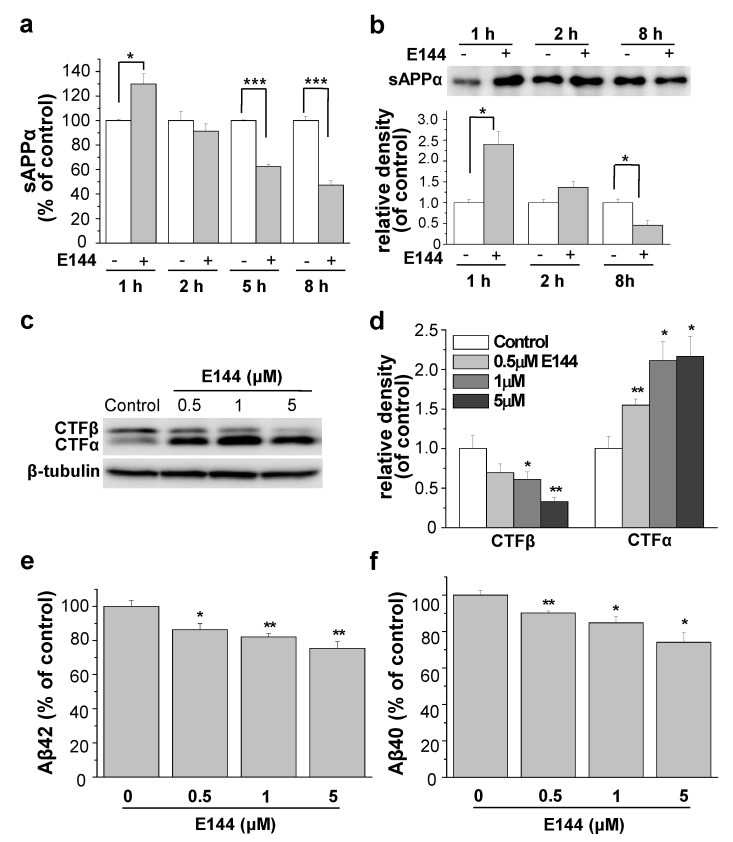
E144 increased the secretion of sAPPα and decreased Aβ. (**a**) APPsw-transfected HeLa cells were incubated with 1 μM E144 for 1, 2, 5, or 8 h. The level of sAPPα in conditioned media was measured using ELISA. The level of sAPPα was significantly increased after incubating with E144 for 1 h (*n* = 6). (**b**) Cells were incubated with 1 μM E144 for various time periods. Conditioned media were incubated with APP antibody against N-terminus, followed by immunoprecipitation with Protein G Agarose. The secreted level of sAPPα was detected using Western blot with the sAPPα antibody. E144 significantly increased the level of sAPPα after incubating with E144 for 1 h (*n* = 5). (**c**) Cells were incubated with 0.5, 1, or 5 μM E144 for 1 h. The level of CTF was detected from cell lysates using Western blot analysis. β-Tubulin was used to confirm the amount of proteins loaded (*n* = 4). The full image of the Western blot is shown in Appendix A. (**d**) Bars correspond to the densitometric analysis of levels of CTFs (*n* = 4). (**e**,**f**) Cells were incubated with E144 for 1 h. Levels of Aβ42 and Aβ40 in conditioned media were measured using ELISA. E144 decreased both Aβ42 and Aβ40 levels in a dose-dependent manner (*n* = 4). * *p* < 0.05; ** *p* < 0.01; *** *p* < 0.001.

**Figure 3 molecules-25-00646-f003:**
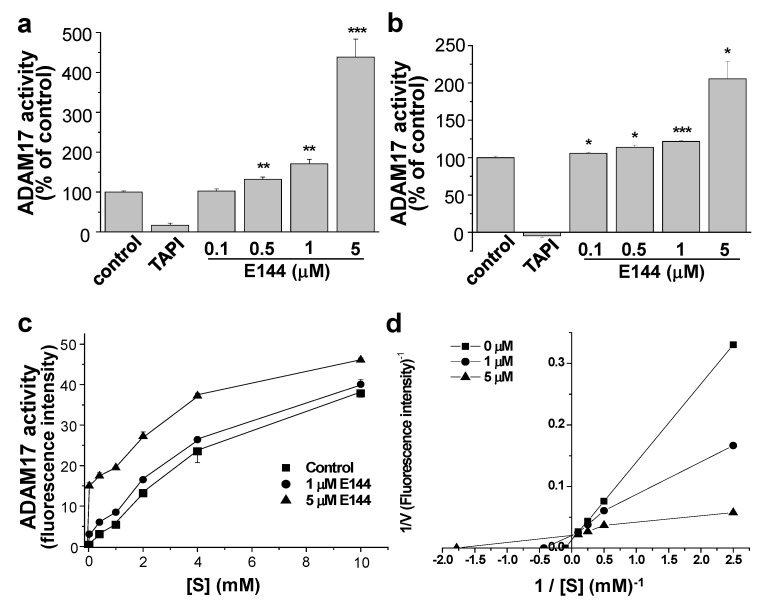
E144 activated A Disintegrin and Metalloproteinase (ADAM)17. (**a**) E144 activated ADAM17 in a cell-based assay. APPsw-transfected HeLa cell lysate was used to measure ADAM17 activity as described in the Materials and Methods. A fluorogenic ADAM17 substrate, Mca-KPLGL-Dpa-AR, was incubated in the presence of 0.1, 0.5, 1, or 5 μM E144 (*n* = 6). ADAM17 activity was inhibited by 50 μM TAPI-1. (**b**) E144 increased ADAM17 activity in a cell-free assay. Human recombinant ADAM17 was used to measure ADAM17 activity in the presence of 0.1, 0.5, 1, or 5 μM E144 (*n* = 4). TAPI-1 at 50 μM inhibited ADAM17 activity. (**c**,**d**) Human recombinant ADAM17 (200 ng/mL) was incubated with 0.02, 0.4, 1, 2, 4, or 10 mM substrate in the presence (1 or 5 μM) or absence of E144 (*n* = 4). After incubation at 37 °C for 5 h, fluorescence was measured. Plots of 1/V versus 1/[S] were fitted by Lineweaver–Burk plot analysis. * *p* < 0.05; ** *p* < 0.01; *** *p* < 0.001.

**Figure 4 molecules-25-00646-f004:**
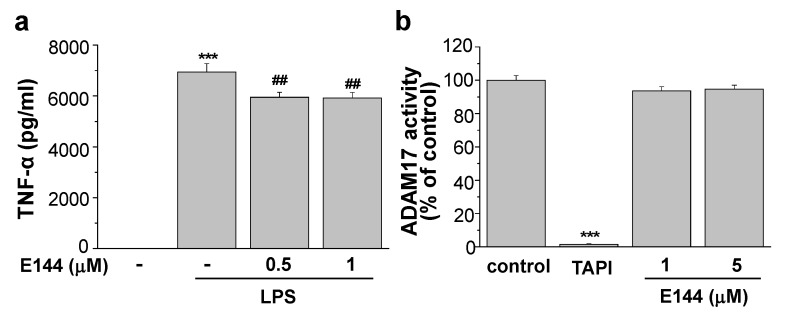
E144 activated ADAM17 in a substrate-specific manner. (**a**) BV-2 cells were incubated with 1 μg/mL LPS for 3 h, and the culture medium was exchanged with a fresh one. Cells were then treated with 0.5 or 1 μM E144 in the presence of LPS for 1 h. Levels of TNFα were measured in conditioned media using ELISA (*n* = 5). (**b**) ADAM17 activity was measured using an assay kit. Human recombinant ADAM17 was incubated for 40 min with 1 or 5 μM E144 in the presence of ADAM17 substrate, QXL520/5-FAM (*n* = 5). TAPI-1 at 50 μM inhibited ADAM17 activity. *** *p* < 0.001 compared to the control group; ^##^
*p* < 0.01 compared to LPS-induced control group.

**Figure 5 molecules-25-00646-f005:**
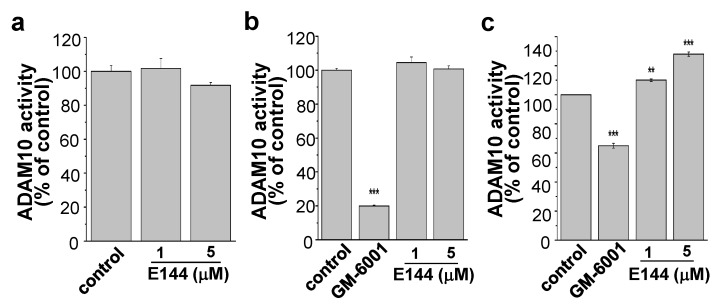
E144 also activated ADAM10 in a substrate-specific manner. (**a**) E144 did not affect ADAM10 activity in the cell-based assay. Lysate of APPsw-transfected HeLa cells was incubated for 40 min with 1 or 5 μM E144 in the presence of ADAM10 substrate, 5-FAM/QXL520 (*n* = 4). (**b**) E144 did not affect ADAM10 activity in a cell-free assay using human recombinant ADAM10 (*n* = 4). (**c**) ADAM10 activity was measured using substrate, Mca-KPLGL-Dpa-AR. Human recombinant ADAM10 was incubated with 1 or 5 μM E144 for 1 h (*n* = 4). ** *p* < 0.01; *** *p* < 0.001.

**Figure 6 molecules-25-00646-f006:**
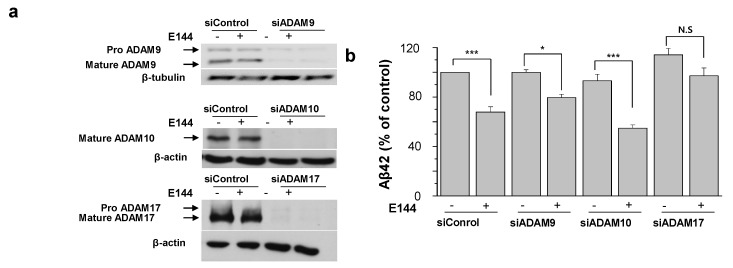
Suppression of ADAM17 expression prevented the effect of E144 on Aβ42 production. APPsw-transfected HeLa cells were treated either with siControl or with siRNA against ADAM9, ADAM10, ADAM17 for 72 h. (**a**) Representative Western blot image showed that the expressions of all three enzymes were suppressed by over 90% in our experimental siRNA conditions (*n* = 6). β-Tubulin and β-actin were used as loading controls. Full images of Western blot are shown in Appendix A. (**b**) At 72 h post-siRNA ADAMs, cells were washed and incubated with 1 μM E144 for 1 h. Levels of Aβ42 in the conditioned media were analyzed using specific ELISA (*n* = 6). Left bars are without E144 in the presence of different siRNAs. * *p* < 0.05; *** *p* < 0.001.s

**Figure 7 molecules-25-00646-f007:**
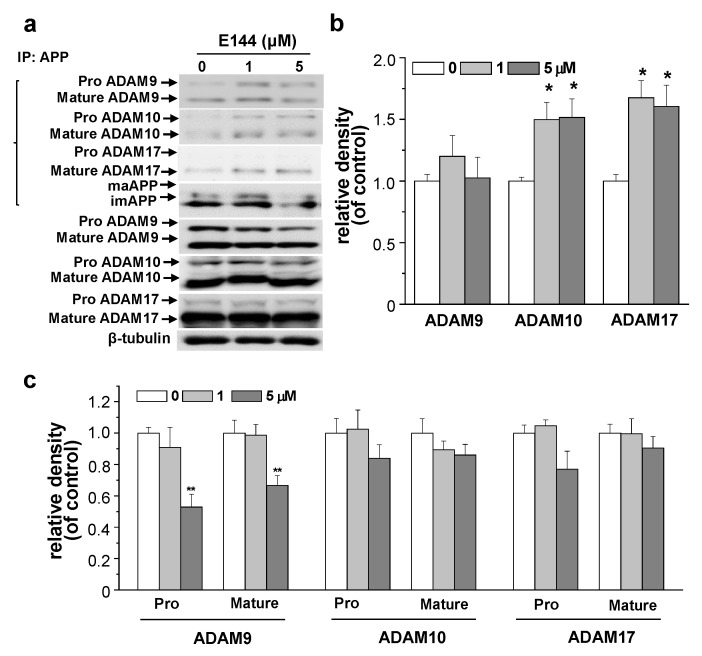
E144 increased the interaction between APP and ADAMs. APPsw-transfected HeLa cells were incubated with indicated concentrations of E144 for 1 h. (**a**) *Upper panel* (IP). Cell lysates were co-immunoprecipitated with APP antibody, followed by Western blot with ADAM9, ADAM10, ADAM17, and APP antibodies. *Lower panel.* Cell lysates were obtained to detect the expression level of ADAM9, ADAM10, and ADAM17. β-Tubulin was used to confirm the amount of proteins loaded. Full images of Western blot are shown in Appendix A. (**b**) Densitometric analysis of Western bands in the *upper panel* (IP) in (a) (*n* = 4). (**c**) ADAM9, ADAM10, and ADAM17 levels were quantified by performing densitometric analysis of bands in the *lower panel* in (**a**) (*n* = 4). * *p* < 0.05; **, *p* < 0.01.

**Figure 8 molecules-25-00646-f008:**
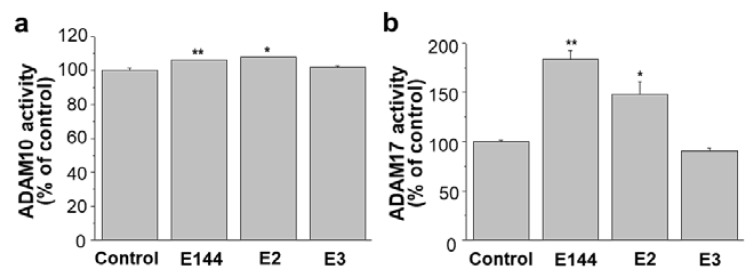
E2 also increases ADAM10 and ADAM17 activities. (**a**) Human recombinant ADAM10 was incubated for 1 h with substrate, Mca-KPLGL-Dpa-AR. The presence of 1 μM E144 and Narciclasine (E2) increased ADAM10 activity (*n* = 3). The presence of 7-Deoxynarciclasine (E3) was without effect. (**b**) APP-transfected HeLa cell lysate was used to measure ADAM17 activity as described in the Materials and Methods. A fluorogenic ADAM17 substrate was incubated in the presence of 1 μM E144, E2, and E3. ADAM17 activity was significantly activated by 1 μM E144 and E2 (*n* = 4). * *p* < 0.05; **, *p* < 0.01.

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
