# Peer review of "Substrate-Specific Activation of α-Secretase by 7-Deoxy-Trans-Dihydronarciclasine Increases Non-Amyloidogenic Processing of β-Amyloid Protein Precursor"

_molecules, 2020, doi:10.3390/molecules25030646_

Round 1
Reviewer 1 Report
This work deals with the "activation of α-secretase by 7- deoxy-trans-dihydronarciclasine increases non-4 amyloidogenic processing of β-amyloid protein precursor". The overall work is interesting and provides to the author many interesting data about the “interaction” of 7- deoxy-trans-dihydronarciclasine with ab peptide and ADAM10/17. Thus, I recommend this work for Molecules.
Author Response
N/A
Reviewer 2 Report
The authors describe a potential molecular basis for the treatment effect of the natural product E144 in animal models. By using a different set of cell-based and in vitro assays, they collect evidence for an activation of ADAM10 and ADAM17 by E144. A few points should be addressed prior to publication:
Major points:
In figures 2a and b, the authors interpret the results of initially increasing (1h) and later decreasing sAPPalpha in terms of activation of ADAMs. As a control, the authors should show also the levels of sAPPbeta (or CTFbeta) as they did in Fig 2c, simply to make sure that the effect is not due to changed APP transport to the membrane. In Figs 3 c and d, the authors analyzed the effect of E144 on ADAM activity in vitro. In lines 180 and 181, they claim that the compound increases Vmax. If all lines intersect on the Y-axis (1/v), however, Vmax remains unchanged upon addition of the activator. Thus, E144 ONLY increases affinity of ADAM to the substrate. The data presented support increased binding of ADAM 10 and ADAM17 to APP (Fig 7). Why did ADAM10 siRNA not counteract the effect of E144 as ADAM17 siRNA did (Figure 6)? The authors might wish to discuss this observation in more detail.Minor points:
Abstract, line 25: …, we further analyzed the activating effect… Intro, line 71: ADAM17-positive Intro, line101: ….as we have previously shown using Western blot analysis. Figure 6b, left bars: siControl Discussion, line 279: levels of Abeta and APP…Author Response
Please see the attachment.

Reviewer 3 Report
The manuscript present interesting and potentially important findings about activity of plant-isolated compounds to the processing of APPsw, with implications to the pathogenesis of Alzheimer's Disease and finding strategies to favorise alpha-secretase activity in APP metabolism, with consequences to find ABETA lowering strategies with fewer side effects.
The manuscript is well written, results are presented clearly and conclusions are supported by them. I have only minor comments/corrections:
Introduction - authors should acknowledge that in the current understanding of Alzheimer's disease pathogenesis, the ABETA is not the only player driving progression of AD. Tau pathology correlates with clinical decline and anti-tau approaches emerge as hope for disease modifying therapy. line 71: correct "ADAM17-positve" to "ADAM17-positive" line 85: correct "treatment of E144" to "treatment with E144" line 365: correct "the detect signals" to "the detected signals" Supplementary Fig. 6 caption: correct "for Fig. 7" to "for Fig. 6" please discuss why recombinant ADAM17 exhibits lower level of activation by E144 (Fig. 3a versus Fig.3b) please describe details of conditioned media concentration approach in paragraph 4.2 Discuss influence of APP Swedish mutation on the processing by alpha secretase, if any Move the results obtained after experiments with E2, E3 compounds from Discussion, lines 326-336 to appropriate parts of Results section.
Author Response
Please see the attachement.

Round 2
Reviewer 2 Report
The authors considered all suggestions, the paper is basically ready for publication. One little remaining issue regards the wording of line 185, which is still confusing. My recommendation would be:
Thus, E144 decreased Michaelis-Menten constant while maximal velocity, Vmax, was apparently not or only moderatly affected
If this Statement is not correct, the authors should think about whether the compound E144 is increasing only affinity of ADAMs to their Substrates.